# Facemask wearing to prevent COVID-19 transmission and associated factors among taxi drivers in Dessie City and Kombolcha Town, Ethiopia

Tarikuwa Natnael[1©], Yeshiwork Alemnew[2], Gete Berihun[1], Masresha Abebe[1], Atsedemariam Andualem[3], Sewunet Ademe[3], Belachew Tegegne[3], Metadel Adane[1©]*

1 Department of Environmental Health, College of Medicine and Health Sciences, Wollo University, Dessie, Ethiopia, 2 Department of Biology, College of Natural Sciences, Wollo University, Dessie, Ethiopia, 3 Department of Nursing, School of Nursing and Midwifery, College of Medicine and Health Sciences, Wollo University, Dessie, Ethiopia

© These authors contributed equally to this work.

* metadel.adane2@gmail.com

**Data Availability Statement:** All relevant data are within the manuscript and its Supporting Information files.

## Abstract

### Background

The World Health Organization (WHO) has pointed out that urban taxi drivers and their passengers are at higher risk of transmitting coronavirus disease 19 (COVID-19) due to frequent contact among many people. Facemask wearing is one of the preventive measures recommended to control the transmission of the virus. A lack of evidence of the proportion of facemask wearing among taxi drivers and associated factors in Ethiopia, including Dessie City and Kombolcha Town, hinders the design of targeted interventions to advocate for facemask use. This study was designed to address this gap.

### Methods

A cross-sectional study was conducted among 417 taxi drivers in Dessie City and Kombolcha Town from July to August, 2020. The study participants were selected using a simple random sampling technique after proportionally allocating the sample size from the total number of taxi drivers working in Dessie City and Kombolcha Town. The data were collected by trained data collectors using a structured questionnaire and an on-the-spot observational checklist. The collected data were checked, coded and entered to EpiData version 4.6 and exported to Statistical Package for Social Sciences (SPSS) version 25.0 for data cleaning and analysis. Bivariate (Crude Odds Ratio [COR]) and multivariable (Adjusted Odds Ratio [AOR]) logistic regression analyses were employed using 95% CI (confidence interval). From bivariate logistic regression analysis, variables with $p$-value < 0.250 were retained into multivariable logistic regression analysis. Then, from the multivariable analysis, variables with $p$-value < 0.050 were declared as factors significantly associated with facemask wearing among taxi drivers in Dessie City and Kombolcha Town.

**Funding:** Wollo University funded this study. The funders had no role in study design, data collection and analysis, decisions to publish, interpretation of the data and preparation of the manuscript for publication.

**Competing interests:** The authors have declared that no competing interests exist.

**Abbreviations:** AOR, adjusted odds ratio; COR, crude odds ratio; CI, confidence interval; COVID-19, Coronavirus disease-19; MERS-COV, Middle East Respiratory Syndrome Coronavirus; SARS-CoV, Severe Acute Respiratory Syndrome Coronavirus; WHO, World Health Organization.

## Main findings

The proportion of taxi drivers who wore a facemask was 54.68% [95%CI: 50.10–59.7%]. The majority (58.3%) of drivers were using cloth facemasks, followed by N95 facemasks (24.5%) and surgical facemasks (17.3%). Out of the total 417 taxi drivers, more than two-thirds (69.8%) of them had a good knowledge about COVID-19 and 67.6% of taxi drivers had a positive attitude towards taking precautions against transmission of COVID-19. Three-fourths (74.1%) of the taxi drivers believed that wearing a facemask could prevent COVID-19. More than half (52.5%) felt discomfort when wearing a facemask. Almost three-fourths (72.2%) of taxi drivers felt that the presence of local government pressure helped them to wear a facemask. We found that marital status [AOR = 3.14, 95%CI: 1.97–5.01], fear of the disease [AOR = 2.1, 95%CI: 1.28–3.47], belief in the effectiveness of a facemask [AOR = 5.6, 95%CI: 3.1–10.16] and feeling government pressure [AOR = 3.6, 95%CI: 2.16–6.13] were factors significantly associated with wearing a facemask.

## Conclusion

We found that the proportion of facemask wearers among taxi drivers was relatively low in Dessie City and Kombolcha Town. In order to increase that number, government bodies should work aggressively to encourage more taxi drivers to wear a facemask. We also recommend that government and non-government organizations work very closely together to implement strategies that promote facemask use, including increasing the availability of inexpensive facemasks, and monitoring and controlling facemask use.

## Introduction

Globally, several different pandemic diseases have been reported in the last two decades. In 2002, Severe Acute Respiratory Syndrome Coronavirus (SARS-CoV) was first reported in Hong Kong and caused 8,422 cases and 916 deaths across 29 countries [1]. Ten years later, in 2012, the Middle East Respiratory Syndrome Coronavirus (MERS-CoV) was first identified in Saudi Arabia and caused severe outbreaks, resulting in 2,494 cases with 858 deaths across 27 countries [2].

Following SARS-CoV and MERS-CoV came Severe Acute Respiratory Syndrome Coronavirus-2 (SARS-CoV-2), formally named as the novel Coronavirus-2019 (nCoV-2019), considered highly pathogenic and a huge outbreak by the World Health Organization (WHO). It is reported to have originated in Hubei province, China during late December 2019 [3]. Since then, the COVID-19 outbreak has become a public health problem around the world [4].

The clinical symptoms of COVID-19 are fever, dry cough, breathing difficulties, headache, pneumonia, and myalgia or fatigue [5, 6]. In addition, severe cases can also lead to cardiac injury, respiratory failure, acute respiratory distress syndrome and ultimately death [7]. In the beginning, COVID-19 was transmitted from animal to human, then human-to-human transmission started through respiratory droplets and body contacts [8, 9]. Another more recent study found transmission also through talking, in particular with louder speech [10]. Transmission can be reduced through preventive measures such as facemasks, hand hygiene, avoidance of public contact, case detection, contact tracing, and quarantines. The COVID-19 can affect all age groups, but the risk of death from the virus has been primarily associated with older ages and chronic diseases [11, 12].

As of 04 October 2020, COVID-19 was responsible for 34.8 million cases and over 1 million deaths globally including over 1 million cases and 26,264 registered deaths in Africa. [13]. It was estimated that COVID-19 would cause 1,222.3 million infections, 22.5 million hospitalizations and 3.3 million deaths in Africa in 2020 in the absence of any intervention [14]. Estimates show that COVID-19 would have resulted in 7 billion infections and 40 million deaths globally this year if no action had been taken [15].

Beyond causing morbidity and mortality, the virus can have a great impact on the economy. A report by the United Nations (UN), has estimated that because of COVID-19 the African continent will face a 1.4 percentage point decline of gross domestic product (GDP) [14]. Since there is currently no cure and vaccines will take a long time to roll out, in order to prevent these diverse impacts, it is essential to focus on prevention measures. Use of facemasks is one of the essential measures to prevent the transmission of COVID-19.

Since the outbreak of COVID-19, the use of facemasks has increased in China and other Asian countries such as South Korea and Japan [16], given that the virus can be transmitted through respiratory droplets even from asymptomatic individuals [17, 18]. In the USA, the rates of COVID-19 cases, hospitalizations, and deaths were reduced dramatically following implementation of facemask wearing recommendations [19, 20]. Even though facemasks have been shown to be a tool to tackle the COVID-19 pandemic, controversy about them continues among some people, and many people, including taxi drivers, do not regularly use a facemask for unknown reasons.

In Ethiopia, the first case of COVID-19 was reported on 13 March 2020, and the number of cases is increasing with improved testing. The WHO reported that the country had 76,988 confirmed cases and 1,207 deaths recorded as of the 04 October 2020 [13]. In response to this, the government of Ethiopia imposed restriction on the movement of taxis across the country, including in Dessie City and Kombolcha Town. When this restriction was found to be difficult, the government policy shifted to allow the movement of taxis while enacting prevention measures such as reducing the number of travellers by half and recommending use of a hand sanitizer and facemask [21, 22]. As of 13 March 2020, the government of Ethiopia imposed the use of facemasks across the country as a means of reducing transmission. These policies show how important the use of facemasks is as a prevention measure for COVID-19.

Despite these efforts, the virus's infection rate is increasing at alarming rate; and it is important, therefore, to strengthen prevention measures. Since taxi drivers have daily close contact with community members while providing their service, assessment of their use of facemask may play a great role in developing strategies to prevent transmission of the virus. The result of a study in six Asian countries revealed that from 103 work-associated COVID-19 cases, 18% happened among drivers and transport workers [23]. Although taxi drivers may be especially vulnerable to the virus, there has been no study to examine the use of prevention measures among this population in Dessie City and Kombolcha Town. Therefore, this study aimed to assess the proportion of facemask wearing and associated factors among taxi drivers in Dessie City and Kombolcha Town in Northeastern Ethiopia.

## Materials and methods

### Study area

Kombolcha Town and Dessie City is found in South Wollo Zone, along the eastern margin of Amhara Regional State in the north central part of Ethiopia, away from 377km and 401km from Addis Ababa, which is a capital city of Ethiopia, respectively. According to the 2007 population and housing census projection, Dessie District had a total population of 212,436 in

2014, of which, 83.6% (177,688) lived in urban *kebeles* (*kebele* is the smallest administrative unit in Ethiopia, each with around 5,000 populations) and 34,748 (16.4%) lived in peri-urban *kebeles* [24] whereas according to Kombolcha town administration report, the population of Kombolcha town was 102,530 in 2020, consisting of 44,305 males and 58,225 females [25].

## Study design and population

A cross-sectional study was conducted from July to August, 2020 in Dessie City and Kombolcha Town in north-eastern Ethiopia. All taxi drivers in Dessie City and Kombolcha Towns were the source populations of the study. The study population was all randomly selected taxi drivers.

## Sample size determination and sampling procedure

Sample size was determined using single population proportion formula [26].

$$n = \frac{(z_{a/2})^2 * p(1-p)}{d^2}$$

Considering the assumptions of:

$Z_{\alpha/2}$ is the standard normal variable value at (1-α) % confidence level (α is 0.05 with 95%CI [confidence interval], $Z_{\alpha/2}$ = 1.96), *p* is an estimate of the proportion of facemask wearing among taxi drivers in Dessie City and Kombolcha Town was 50.0%. A proportion of 50.0% was considered since there had been no previous study conducted in the study area or other similar setting, and *d* margin of error (5.0%). Adjusting for an anticipated 10% non-response rate, the final sample size was determined to be 422.

From a total of eight taxi stands found in Dessie City, five were randomly selected. In addition, of the five taxi stands in Kombolcha Town, four were randomly selected. Last, a proportional allocation of sample size was performed to determine the number of taxi drivers to be included from each taxi stand. Then, simple random sampling technique was employed to select taxi drivers from the respective taxi stands.

## Operational definitions

**Facemask wearing.**   Using a disposable or reusable device that creates a physical barrier between the mouth and nose of the wearer and potential contaminants in the immediate environment.

**Good knowledge.**   Those study participant taxi drivers whose number of correct answers was above or equal to the mean score to 11 knowledge questions about COVID-19.

**Poor knowledge.**   Those study participant taxi drivers whose number of correct answers was below the mean score to 11 knowledge questions about COVID-19.

**Positive attitude.**   Those study participant taxi drivers whose number of correct answers was above or equal to the mean score to 15 attitude questions about taking precautions against transmission of COVID-19.

**Negative attitude.**   Those study participant taxi drivers whose number of correct answers was below the mean score to 15 attitude questions about taking precautions against transmission of COVID-19.

**Proportion of facemask wearing.**   Number of taxi drivers who were wearing a facemask that covered the nose, mouth, lower jaw and facial hair at the time of data collection divided by the total number of study participants.

**Taxi stand.**   Means in Amharic '*fermata*'.

## Data collection and quality assurance

A pretested structured questionnaire adapted from published articles [19, 20, 27–30], WHO report [31], Ethiopian Ministry of Health (EMOH), and Ethiopian Public Health Institute (EPHI) COVID-19 prevention guidelines [32] were used to assess face mask wearing practice among taxi drivers in the study area. The questionnaire was prepared in English, translated to Amharic and then re-translated back to English. The questionnaire had three parts: Part I included questions about socio-demographic and economic factors; Part II covered knowledge- and attitude-related factors and Part III asked about behavioural factors. Two data collectors and one supervisor were recruited for data collection. The data was collected by face-to-face interviews at the work site and by observing utilization of a facemask. The data collectors kept a minimum of 1m distance from the interviewees and used a facemask and hand sanitizer throughout the data collection process.

To ensure the quality of the data, two days of training were given to 2 data collectors and 1 supervisor about the objective of the study, data collection tools, ethical issues and other considerations during data collection such as precaution about COVID-19. The data collectors were also trained about COVID-19 prevention measures. In addition, before the start of the actual data collection the questionnaire was pretested for clarity and consistency on 5% of the sample size from outside the selected taxi stands of both Dessie City and Kombolcha Town. The questionnaire responses were checked daily by the principal investigator and supervisor for completeness and consistency.

To ensure the validity of the survey tool, the questionnaire was developed after reviewing published articles. To check the reliability of the information entered, 10% of the study participants were randomly selected and re-interviewed by another interviewer. The accuracy of data entries was also checked in a randomly selected 10% of the questionnaires.

## Data management and statistical analysis

The collected data was checked, coded and entered into EpiData version 4.6 and exported to SPSS version 25.0 for data cleaning and analysis. Descriptive statistics such as frequencies and percentages for categorical variables and mean with standard deviations for continuous variables were calculated to examine the overall distribution. Taxi drivers wearing a facemask were coded as '1' and those not wearing a facemask were coded '0'. Based on observed facemask use, the proportion of facemask wearing among taxi drivers was estimated by dividing the number of facemask users by the total number of study participants.

Data was analysed using binary logistic regression model with 95% confidence interval (CI). A bivariate logistic regression analysis (Crude Odds Ratio [COR]) and multivariable logistic regression analysis (Adjusted odds ratio [AOR]) were employed. From the bivariate analysis, variables with a $p$-value $< 0.250$ were considered for multivariable logistic regression analysis. From the multivariable logistic regression analysis, variables with a significance level at $p$-value $< 0.050$ were taken as statistically significant and independently associated with facemask wearing among taxi drivers.

The presence of multi-collinearity among independent variables was checked using standard error at the cutoff value of 2. We found a maximum standard error of 1.76, indicated no multi-collinearity. Model fitness was also checked using the Hosmer Lemeshow test [32], finding a $p$-value 0.914, which indicated the model was fit.

## Ethical consideration

Ethical clearance was obtained from the ethical review committee of Wollo University College of Medicine and Health Sciences (Protocol number: CMHS/451/013/2020). Permission to

conduct the study was also obtained from Dessie City and Kombolcha Town Health Bureau. Prior to the data collection, the purpose of the study and the voluntary nature of participation was explained to the participating taxi drivers. Then, verbal consent was obtained from each study participant. Data collectors wore facemasks and maintained social distancing per the WHO guidelines to prevent transmission of the COVID-19. A facemask was provided to any participating taxi driver who did not wear one during the data collection. The confidentiality of the study participants' data was ensured by avoiding possible identifiers such as their names.

## Results

### Socio-demographic and economic characteristics of study participants

Out of 422 taxi drivers in the study, 417 participated, for a response rate of 98.0%. From the total 417 participants, more than half (53.7%) of the taxi drivers were aged between 18 and 29 years and some (15.8%) were >40 years; the mean age was 29.61years (+7.264 SD [standard deviations]). The educational level attained for more than one-third (43.4%) of drivers was secondary level and for some (15.8%) was college level or above. The monthly income of more than one-third (35.3%) of taxi drivers was greater than $142.50 USD (United States Dollars) (Table 1).

### Behavioural factors

Among 417 taxi drivers, nearly half (48.9%) of the drivers felt they were vulnerable to the virus while over half (51.1%) did not feel vulnerability. Almost two-thirds (62.8%) of taxi drivers

**Table 1. Bivariate analysis of socio-demographic and economic factors associated with facemask wearing among taxi drivers in Dessie City and Kombolcha Town, Northeastern Ethiopia, July to August 2020.**

| Variable | Frequency (N = 417) | Facemask wearing | | COR (95% CI) | p-value |
|---|---|---|---|---|---|
| | | Yes | No | | |
| | n(%) | n | n | | |
| **Driver's age (years)***| | | | | |
| 18–29 | 224(53.7) | 120 | 104 | Ref | |
| 30–39 | 127(30.5) | 67 | 60 | 0.97(0.63–1.49) | 0.884 |
| >40 | 66(15.8) | 41 | 25 | 1.42(0.81–2.49) | 0.221 |
| **Driver's education status** | | | | | |
| Primary (up to grade 8) | 170(40.8) | 91 | 79 | Ref | |
| Secondary (from grades 9–12) | 181(43.4) | 104 | 77 | 1.17(0.77–1.78) | 0.461 |
| College or above | 66(15.8) | 33 | 33 | 0.87(0.49–1.53) | 0.630 |
| **Marital status** | | | | | |
| Unmarried | 178(42.7) | 72 | 106 | Ref | |
| Married | 239(57.3) | 156 | 83 | 2.76(1.85–4.13) | < 0.001 |
| **Type of area where grew up** | | | | | |
| **Urban** | 376(90.2) | 207 | 169 | 1.16(0.61–2.22) | 0.643 |
| **Rural** | 41(9.8) | 21 | 20 | Ref | |
| **Monthly income (USD)** | | | | | |
| ≤$142.50 | 270(64.7) | 147 | 123 | Ref | |
| >$142.50 | 147(35.3) | 81 | 66 | 1.03(0.68–1.54) | 0.893 |
| **Household size (persons)** | | | | | |
| ≤5 | 376(90.2) | 207 | 169 | 1.16(0.61–2.22) | 0.647 |
| >5 | 41(9.8) | 21 | 20 | Ref | |

*Mean driver age was 29.61 + 7.264 years; USD, United States Dollars

Ref, reference category; COR, crude odds ratio; CI, confidence interval

feared the virus and more than one-third (37.2%) did not fear the virus. Three-fourths (74.1%) of the taxi drivers believed that wearing a facemask could prevent them from contracting and spreading COVID-19. Nearly one-third (31.9%) of taxi drivers had difficulty in obtaining/accessing a facemask, whereas more than half (52.5%) felt discomfort when wearing a facemask. Almost three-fourths (72.2%) of taxi drivers felt that the presence of local government pressure helped them to wear a facemask (Table 2).

### Knowledge and attitude of taxi drivers towards COVID-19

Out of the total 417 taxi drivers, over two-thirds (69.8%) had good knowledge about COVID-19 whereas almost a third (30.2%) had poor knowledge. With regard to attitude about the pandemic, about two-thirds (67.6%) of taxi drivers had positive attitude and almost one-third (32.4%) had negative attitude towards taking precautions against transmission of COVID-19 (Table 3).

### Proportion of facemask wearing

The proportion of taxi drivers who wore a facemask was 54.68% [95%CI: 50.1–59.7%] and who did not wear facemask 45.32% [95%CI: 40.3–49.9%]. More than half (58.3%) of the drivers wore cloth facemasks, followed by N95 facemasks (24.5%) and surgical facemasks (17.3%). With regard to the disposal of facemasks, the majority (52.0%) of drivers disposed of their facemask in a pit and some (1.7%) practiced open disposal (Fig 1).

### Factors associated with facemask wearing

From multivariable logistic regression analysis, the factors of marital status, fear of COVID-19, belief in the effectiveness of facemasks and presence of government pressure to wear a facemask showed significant association with facemask wearing among taxi drivers (Table 4).

Married individuals were 3.14 times more likely to wear a facemask than unmarried individuals (AOR = 3.14, 95%CI: 1.97–5.01). Fear of the disease also showed significant association with facemask wearing. Individuals who feared the virus were 2.1 times more likely to wear a facemask than individuals who did not fear the disease (AOR = 2.1, 95%CI: 1.28–3.47). On the other hand, individuals who believed in the effectiveness of facemasks were 5.6 times more likely to wear a facemask than those who did not believe in the effectiveness of facemasks (AOR = 5.6, 95%CI: 3.1–10.16). Similarly, drivers who reported the presence of government pressure were 3.6 times more likely to wear a facemask than those who did not (AOR = 3.6, 95%CI: 2.16–6.13) (Table 4).

## Discussion

This cross-sectional study was conducted with the aim of determining the proportion and associated factors of facemask wearing among taxi drivers in urban areas of Dessie and Kombolcha. We found that the proportion of facemask wearing among taxi drivers was 54.68% and that facemask wearing among taxi drivers was significantly associated with marital status, fear of COVID-19, a belief in the effectiveness of facemask wearing and feeling government pressure to wear facemask.

This study found that the proportion of facemask wearing among taxi drivers was relatively low. This finding is in line with a study conducted in China among elementary school students, which found facemask wearing was 51.6% [34]. It is lower than the proportion found in South Korea among the general population, which was 63.2% [35] in Beijing among healthcare workers at 70% [36], in Hong Kong Special Administrative Region (HKSAR) among the

**Table 2. Bivariate analysis of behavior related factors associated with facemask wearing among taxi drivers in Dessie City and Kombolcha Town, Northeastern Ethiopia, July to August 2020.**

| Variable | Frequency (N = 417) | Facemask wearing | | COR (95% CI) | p-value |
|---|---|---|---|---|---|
| | | Yes | No | | |
| | n(%) | n | n | | |
| **Feeling vulnerable to COVID-19** | | | | | |
| Yes | 204(48.9) | 124 | 80 | 1.62(1.10–2.39) | 0.014 |
| No | 213(51.1) | 104 | 109 | Ref | |
| **Knowing individual(s) infected with COVID-19** | | | | | |
| Yes | 27(6.5) | 16 | 11 | 1.22(0.55–2.69) | 0.621 |
| No | 390(93.5) | 212 | 178 | Ref | |
| **Feel fear of COVID-19** | | | | | |
| Yes | 262(62.8) | 173 | 89 | 3.53(2.33–5.36) | < 0.001 |
| No | 155(37.2) | 55 | 100 | Ref | |
| **Worry that Dessie City/Kombolcha Town would become a quarantine city/town** | | | | | |
| Yes | 165(39.6) | 87 | 78 | 0.87(0.59–1.3) | 0.518 |
| No | 252(60.4) | 141 | 111 | Ref | |
| **Believe that wearing facemask could prevent contracting and spreading of COVID-19** | | | | | |
| Yes | 309(74.1) | 206 | 103 | 7.82(4.63–13.21) | < 0.001 |
| No | 108(25.9) | 22 | 86 | Ref | |
| **Have difficulty in obtaining/accessing facemask** | | | | | |
| Yes | 133(31.9) | 67 | 66 | Ref | |
| No | 284(68.1) | 161 | 123 | 1.28(0.85–1.94) | 0.231 |
| **Feel discomfort when wearing a facemask** | | | | | |
| Yes | 219(52.5) | 116 | 103 | Ref | |
| No | 198(47.5) | 112 | 86 | 1.16(0.78–1.70) | 0.460 |
| **Feel presence of government pressure to wear facemask** | | | | | |
| Yes | 301(72.2) | 193 | 108 | 4.14(2.61–6.56) | < 0.001 |
| No | 116(27.8) | 35 | 81 | Ref | |
| **Feel family members' encouragement to wear facemask** | | | | | |
| Yes | 210(50.4) | 132 | 78 | 1.96(1.32–2.89) | 0.001 |
| No | 207(49.6) | 96 | 111 | Ref | |
| **Feel passengers' encouragement to wear facemask** | | | | | |
| Yes | 160(38.4) | 84 | 76 | 0.87(0.58–1.29) | 0.481 |
| No | 257(61.6) | 144 | 113 | Ref | |
| **Workplace acceptance of wearing facemask** | | | | | |
| Yes | 263(63.1) | 140 | 123 | 0.85(0.57–1.27) | 0.442 |
| No | 154(36.9) | 88 | 66 | Ref | |
| **Knowing that COVID-19 can be a fatal disease** | | | | | |
| Yes | 385(92.3) | 209 | 176 | 0.81(0.39–1.69) | 0.581 |
| No | 32(7.7) | 19 | 13 | Ref | |

Ref, reference category; COR, crude odds ratio; CI, confidence interval

community, 96.6% [37], in Wuhan, Hubei, China among market visitors during COVID-19, 99.7% [38], among individuals visiting hospital during COVID-19, 96.9% [39], among pedestrians in several regions across Hong Kong, 94.8% [40], or among adult residents in Hong Kong, 61.2% [28] but, higher than found in USA among shoppers at 41% [29], in Hong Kong among the community, 21% [41], in Japan among adults, 38% [42] or in another study done

**Table 3. Bivariate analysis of knowledge and attitude associated with facemask wearing among taxi drivers in Dessie City and Kombolcha Town, Northeastern Ethiopia, July to August 2020.**

| Variable | Frequency (N = 417) | Facemask wearing | | COR (95%CI) | p-value |
|---|---|---|---|---|---|
| | | Yes | No | | |
| | n(%) | n | n | | |
| **Knowledge about COVID-19** | | | | | |
| Good | 291(69.8) | 156 | 135 | 1.15(0.75–1.75) | 0.510 |
| Poor | 126(30.2) | 72 | 54 | Ref | |
| **Attitude towards taking precautions against transmission of COVID-19** | | | | | |
| Positive | 282(67.6) | 160 | 122 | 1.29(0.85–1.95) | 0.220 |
| Negative | 135(32.4) | 68 | 67 | Ref | |

Ref, reference category; COR, crude odds ratio; CI, confidence interval

in Korea among adults, 15.5% [43]. The higher proportion of facemask wearing in this study compared to those other areas might have been due to the pressure from the government to wear a facemask. In addition to this, the difference may be due to factors such as different study area and study period.

In our study, the majority of drivers were using cloth facemasks, followed by N95 facemasks and surgical facemasks. In contrast to our findings, the majority of the participants were wearing medical facemasks in China among market visitors at 78.8% [38], in Malaysia among hospital visitors 72.0% [38] and in Hong Kong among pedestrians 83.7% [40]. A possible explanation for the lower proportion of medical masks in our study area might be the high cost of medical masks in our study area. In this study, the majority of study participants wore the facemask properly covering mouth, nose, lower jaw and facial hair; others wore their

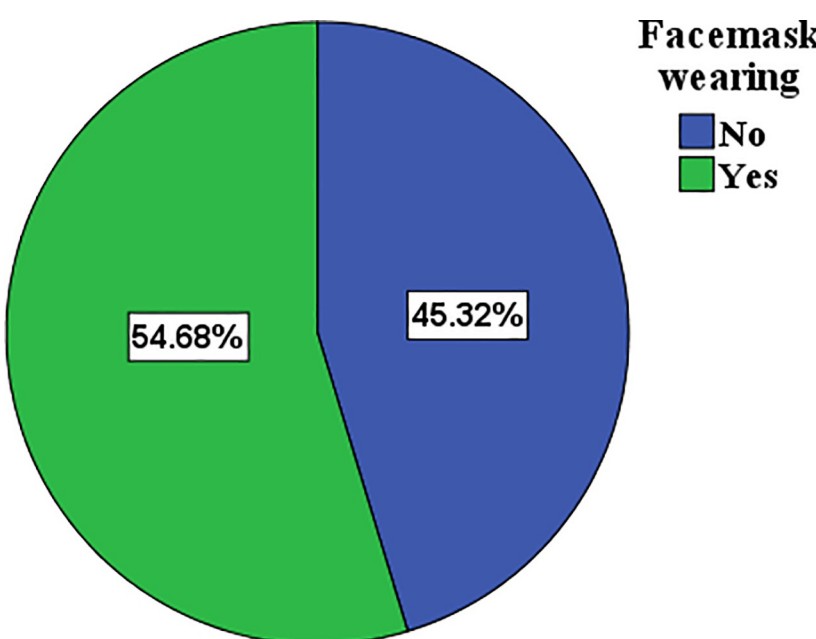

**Fig 1. Proportion of facemask wearing among taxi drivers in Dessie City and Kombolcha Town, Northeastern Ethiopia, July to August 2020.**

**Table 4. Multivariable analysis of factors associated with facemask wearing among taxi drivers in Dessie City and Kombolcha Town, Northeastern Ethiopia, July to August 2020.**

| Variable | Facemask wearing | | AOR (95% CI) | p-value |
|---|---|---|---|---|
| | Yes | No | | |
| | n | n | | |
| **Marital status** | | | | |
| Married | 156 | 83 | 3.14(1.97–5.01) | < 0.001 |
| Unmarried | 72 | 106 | Ref | |
| **Feel fear of COVID_19** | | | | |
| Yes | 173 | 89 | 2.12(1.28–3.47) | 0.003 |
| No | 55 | 100 | Ref | |
| **Believe in the effectiveness of facemask-wearing** | | | | |
| Yes | 206 | 103 | 5.6(3.10–10.16) | < 0.001 |
| No | 22 | 86 | Ref | |
| **Feel presence of government pressure to wear facemask** | | | | |
| Yes | 241 | 68 | 3.64(2.16–6.13) | < 0.001 |
| No | 60 | 48 | Ref | |

Ref, reference category; AOR, adjusted odds ratio; CI, confidence interval

facemasks improperly. Improper facemask wearing practices was also observed in Hong Kong where 13% wore them incorrectly; of these, 35.5% were worn 'inside-out' or 'upside-down' and 42.5% exposed the nostrils or mouth [40].

Our study revealed that the proportion of facemask wearing was higher among married taxi drivers than among unmarried taxi drivers. This difference in practice leads the unmarried individuals to be more vulnerable to COVID-19. This finding is similar to that of a study conducted in Hong Kong [28]. The finding is also supported by the study by Taylor *et al*. [44], which found that unmarried individuals showed lower compliance compared with those who were married. The reason for marital status being a factor for the wearing of facemask might be the fact that married individuals feel more responsible for the health of their families as well as their own compared to unmarried individuals.

This study showed that individuals who had a belief in the effectiveness of wearing facemasks were more likely to wear a facemask. Consistent with our finding, several studies revealed that believing in the effectiveness of a facemask can play a great role in facemask wearing behaviour [28, 41, 45–47]. This may be due to the human behaviour of acting in accordance with one's belief. While measures such as lockdown and self-isolation play a major role in response to spread of the virus, causing 66% reduction in cases and 66% increase in recovered cases [48], wearing a facemask can be an effective method to prevent the spread of the disease in places where it is difficult to keep the recommended social distance. The result of a systematic review on uptake and effectiveness of facemasks at mass gatherings also revealed that facemasks are effective against respiratory infections [49].

This study showed that the proportion of facemask wearing was higher among taxi drivers who feared COVID-19. Individuals who feared the disease were more likely to wear a facemask than those who did not fear the disease. This finding is similar to that of a study conducted in Hong Kong [28]. The finding is also consistent with a recent study conducted among Turkish adults during the COVID-19 outbreak [50]. The possible reason is that knowledge of the serious nature of the epidemic led the drivers to fear the disease and to want to take protective measures against it, including the wearing of a facemask.

This study also found that drivers who reported that there was pressure from the government to wear facemasks were more likely to wear one than others who did not report that pressure. This result is supported by a recent study in the USA where facemask wearing dramatically increased from 41% to 90% among some groups following government pressure [29]. This finding is also similar to studies conducted in Hong Kong [28], in Japan [51] and in Mexico City public transportation during the influenza outbreak [52]. In Ethiopia, wearing a facemask is mandatory for taxi drivers with a fine of 1000 Ethiopian birrs for driving without one on. Even so, it is mostly practiced in places where there is a regular and strong control by the traffic police.

The study has some limitations: due to the scarcity of literature on the proportion of facemask wearers among taxi drivers, discussion was made on the basis of findings from different source populations and studies conducted in other areas. Most of the data was also collected using self-report by taxi drivers, which may underestimate or overestimate the findings of the independent variables. Despite these limitations, the study provides new insight into the extent of facemask wearing among taxi drivers during the COVID-19 pandemic in Dessie City and Kombolcha Town, Ethiopia. Further study that focused on nationwide mask practices is highly encouraged to provide more representative evidence for Ethiopian taxi drivers.

## Conclusion

This study revealed that the proportion of facemask wearing among taxi drivers in the study area was relatively low. The main factors that were significantly associated with facemask wearing were marital status, fear of COVID-19, belief in the effectiveness of facemasks and presence of government pressure to wear facemask. We recommend that Dessie City and Kombolcha Town Health Offices and other concerned government bodies work aggressively to pressure taxi drivers to wear facemasks. We also recommend that government and non-government organizations work very closely to implement strategies that promote facemask use, including increasing the availability of inexpensive facemasks, and monitoring and controlling facemask use. It is also strongly advised that Dessie City and Kombolcha Town traffic police should monitor and control the proper and regular wear of facemasks by taxi drivers.

## Supporting information

**S1 Questionnaire. English version of the questionnaire.** Survey of facemask wearing to prevent COVID-19 transmission and associated factors among taxi drivers in Dessie City and Kombolcha Town, Ethiopia.
(DOCX)

**S2 Questionnaire. Amharic (local language) version of the questionnaire.** Survey of facemask wearing to prevent COVID-19 transmission and associated factors among taxi drivers in Dessie City and Kombolcha Town, Ethiopia.
(DOCX)

**S1 Date set. Minimal data for facemask wearing to prevent COVID-19 transmission and associated factors among taxi drivers in Dessie City and Kombolcha Town, Ethiopia.**
(XLSX)

## Acknowledgments

We acknowledge Dessie City and Kombolcha Town Health Bureaus for their cooperation during this study by providing all the necessary information. We also acknowledge taxi drivers'

associations of Dessie City for their support. We would also like to extend our special thanks to data collectors, the supervisor, and the participating taxi drivers for their valuable contributions to the study. Our sincere thanks are also extended to Ing. Selamu Temesgen from Wollo University, Kombolcha Institute of Technology for his support during the questionnaire development. We also thank Lisa Penttila for language editing of the manuscript.

## Author Contributions

**Conceptualization:** Tarikuwa Natnael, Yeshiwork Alemnew, Gete Berihun, Masresha Abebe, Atsedemariam Andualem, Sewunet Ademe, Belachew Tegegne, Metadel Adane.

**Data curation:** Tarikuwa Natnael, Metadel Adane.

**Formal analysis:** Tarikuwa Natnael, Metadel Adane.

**Funding acquisition:** Tarikuwa Natnael, Yeshiwork Alemnew.

**Investigation:** Tarikuwa Natnael, Yeshiwork Alemnew, Gete Berihun, Masresha Abebe, Atsedemariam Andualem, Sewunet Ademe, Belachew Tegegne, Metadel Adane.

**Methodology:** Tarikuwa Natnael, Yeshiwork Alemnew, Gete Berihun, Masresha Abebe, Atsedemariam Andualem, Sewunet Ademe, Belachew Tegegne, Metadel Adane.

**Project administration:** Tarikuwa Natnael, Yeshiwork Alemnew, Gete Berihun, Masresha Abebe, Atsedemariam Andualem, Sewunet Ademe, Belachew Tegegne, Metadel Adane.

**Resources:** Tarikuwa Natnael, Yeshiwork Alemnew, Gete Berihun, Masresha Abebe, Atsedemariam Andualem, Sewunet Ademe, Belachew Tegegne, Metadel Adane.

**Software:** Tarikuwa Natnael, Yeshiwork Alemnew, Gete Berihun, Masresha Abebe, Atsedemariam Andualem, Sewunet Ademe, Belachew Tegegne, Metadel Adane.

**Supervision:** Tarikuwa Natnael, Yeshiwork Alemnew, Gete Berihun, Masresha Abebe, Atsedemariam Andualem, Sewunet Ademe, Belachew Tegegne, Metadel Adane.

**Validation:** Tarikuwa Natnael, Yeshiwork Alemnew, Gete Berihun, Masresha Abebe, Atsedemariam Andualem, Sewunet Ademe, Belachew Tegegne, Metadel Adane.

**Visualization:** Tarikuwa Natnael, Yeshiwork Alemnew, Gete Berihun, Masresha Abebe, Atsedemariam Andualem, Sewunet Ademe, Belachew Tegegne, Metadel Adane.

**Writing – original draft:** Tarikuwa Natnael, Metadel Adane.

**Writing – review & editing:** Metadel Adane.

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
