## [Decision Letter · Decision Letter 0]

21 Jan 2021

PONE-D-20-33269

Proportion of facemask wearing to prevent COVID-19 and associated factors among taxi drivers in urban areas of Dessie City and Kombolcha Towns in Ethiopia

PLOS ONE

Dear Dr. Adane (PhD),

Thank you for submitting your manuscript to PLOS ONE. After careful consideration, we feel that it has merit but does not fully meet PLOS ONE’s publication criteria as it currently stands. Therefore, we invite you to submit a revised version of the manuscript that addresses the points raised during the review process.

Please attend to all the comments and concerns that have been raised by the reviewer in addition to these I have added below:

1. The Title: I suggest that you amend  the title of the manuscript to read "Proportion of taxi drivers wearing face mask to prevent COVID-19 transmission and associated factors  in urban areas of Dessie City and Kombolcha Towns in Ethiopia".

2. The English language in the manuscripts makes reading it not an easy undertaking. I suggest that you seek the services of a native English speaker or commercial service before re-submitting. 

3. All Tax drivers were eligible for inclusion in the study. Therefore, there is no need for inclusion and Exclusion criteria.

4. Give a reference for the formula used for sample size calculation

5. What is a fermata? Please try to use widely accepted English terms.

6. Give a reference for the publication where your questionnaire was used.

7. Binary logistic regression is different from bivariate analysis. Be clear on how you use these two of terms. Was the multivariate analysis used different from the binary logistic regression and if yes, what was the multivariate model used?

8. Write p-values to three decimal places

9. Please note that some of the references you are using to compare your findings to those of other countries/ studies refer to different segments of populations and diseases (which may be perceived differently by the study populations). As such you have to take precautions when making such comparisons, instead of making general statements.

10. For Table 2, is it divided into several sub-tables or it was just a numbering error?

We look forward to receiving your revised manuscript.

Kind regards,

Martin Chtolongo Simuunza, PhD

Academic Editor

PLOS ONE

Journal Requirements:

2. Please include additional information regarding the survey or questionnaire used in the study and ensure that you have provided sufficient details that others could replicate the analyses. For instance, if you developed a questionnaire as part of this study and it is not under a copyright more restrictive than CC-BY, please include a copy, in both the original language and English, as Supporting Information, or include a citation if it has been published previously.

3. In statistical methods, please clarify whether you corrected for multiple comparisons.

4. Please include a copy of Table 4 which you refer to in your text on page 12.

5.We suggest you thoroughly copyedit your manuscript for language usage, spelling, and grammar. If you do not know anyone who can help you do this, you may wish to consider employing a professional scientific editing service.  

Reviewers' comments:

Reviewer's Responses to Questions

**Comments to the Author**

1. Is the manuscript technically sound, and do the data support the conclusions?

Reviewer #1: Yes

2. Has the statistical analysis been performed appropriately and rigorously? 

Reviewer #1: Yes

3. Have the authors made all data underlying the findings in their manuscript fully available?

Reviewer #1: Yes

4. Is the manuscript presented in an intelligible fashion and written in standard English?

Reviewer #1: Yes

5. Review Comments to the Author

Reviewer #1: Review report (Habtamu Tolera)

The manuscript reports the findings regarding “Proportion of facemask wearing to prevent COVID-19 and associated factors among taxi drivers in urban areas of Dessie

City and Kombolcha Towns in Ethiopia”. This kind of research topic is much relevant, timely and worrying much in the context of LMICs like Ethiopia and the pandemic today’s world as a whole. The analysis was well done. However, I have some concerns (on heading, abstract, methodology, and result sections) which needed be addressed to get published in PLoS ONE. Author(s) can also refer to the attached pdf file.

• Authors did not give line pages to make review work ease. Why??

Heading of the study:

• Authors need to rewrite the word “Facemask” as two words (face mark), not as one word as author(s) described it in the title). keep the consistency across the manuscript.

• Authors are calling Dessie as a city. I suggest they need to use both Dessie and Komblcha as a town because both are really towns. In the current stastus, Dessie and her sister other urban centers like Debre markos, Nekemte, etc do not qualify for the standard of to be named as city (such hierarchy). Even regional capitals do not officially maintain such a rank except Addis.

Abstract

• The last sentence of the “Background” section under the Abstract is too long sentence and lacks language clarity. It has been rewritten.

• The “Conclusion” section under the Abstract which was submitted online in the submission system (first page of the submission system) was not complete. Authors need to recheck.

Introduction:

• PLoS ONe use Vancouver styles of citations, use series Arabic numerals or number citations inside a rectangular shaped bracket, [ ]. For further insight, get and read in-text-citation styles and bibliography writing from authors guidelines of the Journal. So, they need to rework accordingly.

• Authors try to cite or use international and national figures (reports) in the Introduction section to describe the background picture of the study topic. In this regard it is well done. However, I have some concerns.The introduction part needs more detail review of relevant empirical works the world experienced on pandemic diseases like COVID-19 in different time. I suggest that authors improve the description in this regard to provide more justification for their study, specifically, they should expand upon the knowledge gap being filled citing adequate related works.

Methodology:

• Under study design and study areas, Why opt for Dessie and Kombolcha towns than other higher urban center in the region? Here, author(s) needs to justify their rationale. In the same section, authors stated something contradicting with what they reported in the " Abstract" section under the "Method". There in the Abstract section and in each Table caption, they mentioned that the study was conducted June to August 2020 whereas here they stated it was conducted from July to August. Keep the consistency across the manuscript. Authors also said like this, the town lies on the intersection of 11°8′N 39°38′E. this shows a point, not an area. I hope, Dessie is a polygon made up of two or more kebeles. They need to use both the latitudinal and longitudinal extents of the two urban centers independently to depict their astronomical location because these towns are here polygons (areas) since they are study settings, should not be a point in such a study. I also advise authors to show the location of the study areas mapping the two study settings using GIS environment.

• Under Methodological part, there is a sub-section called “Outcome variable measurement and estimation of proportion of facemask wearing” I have found this miss placed sub heading, no need of stating it here as a method of data analysis because it is expected (even you described it in your method of Data magt and analysis sub-section) that authors undertake descriptive statistics like proportion, cross tabs, and like that below. avoid such repetition of data analysis from here. So, advise author(s) move it (this sub-section) immediate to the "Result" section and reportedly presents the two prevalence, wearing and not wearing face cover. authors also gave definition for Proportion of facemask wearing under Operational definitions below. Likewise edit the coding of outcome variables like this, … otherwise, those who do not wear the face mask was coded 0.

Ethics approval and consent to participate

• Cite a protocol code that verified approval IRB of the WU/college

In text citation

• PLoS ONe use Vancouver styles of citations, use series Arabic numerals or number citations inside a rectangular shaped bracket. read in-text-citation and bibliography citation styles of authors guideline. So, rework accordingly

.

End

6. PLOS authors have the option to publish the peer review history of their article (what does this mean?). If published, this will include your full peer review and any attached files.

Reviewer #1: **Yes: **Habtamu Tolera Deressa

---

## [Author Response · Author response to Decision Letter 0]

24 Jan 2021

Date: 24 January 2021

Manuscript ID: PONE-D-20-33269R1

Proportion of taxi drivers wearing facemask to prevent COVID-19 transmission and associated factors in urban areas of Dessie City and Kombolcha Town in Ethiopia

Corresponding authors: Metadel Adane (PhD)

Dear Dr Martin Chtolongo Simuunza (PhD)

Academic Editor

PLOS ONE

Thank you for your letter dated January 21, 2021 with a decision of minor revision needed. We were pleased to know that our manuscript was considered potentially acceptable for publication in PLoS ONE, subject to adequate revision as requested by the reviewers, academic editors and the journals. Based on the instructions provided in your letter, we uploaded the file of the rebuttal letter; the marked up copy of the revised manuscript highlighting the changes made in the original submitted version and the clean copy of the revised manuscript. 

We have revised the manuscript by modifying the abstract, introduction, methods, results, discussion and other sections, based on the comments made by the reviewers and using the journal guidelines. Accordingly, we have marked in red color all the changes made during the revision process. Appended to this letter is our point-by-point response (rebuttal letter) to the comments made by the reviewers. 

We agree with almost all the comments/questions raised by the reviewers and provided justification for disagreeing with some of them. We would like to take this opportunity to express our thanks to the reviewers for their valuable comments and to thank you for allowing us to resubmit a revision of the manuscript. 

I hope that the revised manuscript is accepted for publication in PLoS ONE. 

Sincerely yours,

Metadel Adane (PhD) 

Response to the Journal Requirements Questions 

 Question #1: Please ensure that your manuscript meets PLOS ONE's style requirements, including those for file naming.

Response: Thank you for this remark. We re-formatted the revised manuscript using the PLoS ONE format guidelines. The whole content of the manuscript, including the abstract, introduction, methods, discussion and reference are formatted using the guidelines (please see the revised version for each section).

Question #2. Please include additional information regarding the survey or questionnaire used in the study and ensure that you have provided sufficient details that others could replicate the analyses. For instance, if you developed a questionnaire as part of this study and it is not under a copyright more restrictive than CC-BY, please include a copy, in both the original language and English, as Supporting Information.

Response: We provided the questionnaire in original language (Amharic) and English version as supporting information S1 and S2. 

Question #3. - In statistical methods, please clarify whether you corrected for multiple comparisons.

Response: As you can see in Tables 1-4, first we did bivaraible analysis for each independent variable with the outcome variable, then we did adjusted analysis (multiple comparisons) for controlling the confounders. We already mentioned this procedure at data analysis section. 

Question#4. Please include a copy of Table 4 which you refer to in your text on page 12.

Response: We putted Table 4 and other Tables below the references. Tables 1-4 also cited in the text. Please see from pages 24 to 28. 

Question #5. We suggest you thoroughly copyedit your manuscript for language usage, spelling, and grammar. If you do not know anyone who can help you do this, you may wish to consider employing a professional scientific editing service. 

Response: Response: We edited the manuscript using a friend who is a native speaker, whose name is Lisa Penttila. We acknowledge Lisa for her language editing of the manuscript in the acknowledgment section. See in page 16 from lines 374 to 375.

Reviewer I: Comments by the editor 

1. The Title: I suggest that you amend the title of the manuscript to read "Proportion of taxi drivers wearing face mask to prevent COVID-19 transmission and associated factors in urban areas of Dessie City and Kombolcha Towns in Ethiopia".

Response: We thank you for your valuable comment for amending the title. We accepted your suggested title and please see the revised title. The updated title read as “Proportion of taxi drivers wearing facemask to prevent COVID-19 transmission and associated factors in urban areas of Dessie City and Kombolcha Towns in Ethiopia

2. The English language in the manuscripts makes reading it not an easy undertaking. I suggest that you seek the services of a native English speaker or commercial service before re-submitting. 

Response: We edited the manuscript using a friend who is a native speaker, whose name is Lisa Penttila. We acknowledge Lisa for her language editing of the manuscript in the acknowledgment section. See in page 16 from lines 374 to 375.

3. All Tax drivers were eligible for inclusion in the study. Therefore, there is no need for inclusion and Exclusion criteria. 

Response: Thank you for this pertinent feedback and we deleted the inclusion and exclusion criteria. 

4. Give a reference for the formula used for sample size calculation

Response: We cited the source of the formula (See the sample size determination in page 7 in lines 149). 

5. What is a fermata? Please try to use widely accepted English terms.

Response: We used a more common term in English, which means taxi stand. In Ethiopia, Taxi stand- means in Amharic ‘fermata’. We defined taxi stand in the operational definition section to make clear for Ethiopian readers and taxi stand for international readers (see in page 8 in line 178). 

6. Give a reference for the publication where your questionnaire was used.

Response: We cited the source of the reference, please see the updated version of the manuscript in page 8 from lines 180 to 183. Thank you for this pertinent comment. 

7. Binary logistic regression is different from bivariate analysis. Be clear on how you use these two of terms. Was the multivariate analysis used different from the binary logistic regression and if yes, what was the multivariate model used?

Response: Binary logistic regression model, which is commonly also called logistic regression is a model used to identify factors for the binary outcome variable (in our case facemask wearing and not facemask wearing). Thus, we used to steps of analysis, the first was bivariable analysis (analysis of one independent variable with outcome without controlling the confounders). Those variables that has a p-value of less than 0.25 was included to the multivariable analysis, which is also commonly called multivariable logistic regression analysis, a model used for adjusted analysis to control confounders. We revised the paper in the way to avoid confusion (Please see the updated data analysis from the abstract in page 3 and page 9 to 10 form lines 208 to 222). 

8. Write p-values to three decimal places

Response: We wrote the p-value for 3 digits. Please see Table 4 and throughout the paper. 

9. Please note that some of the references you are using to compare your findings to those of other countries/ studies refer to different segments of populations and diseases (which may be perceived differently by the study populations). As such you have to take precautions when making such comparisons, instead of making general statements. 

Response: Thank you for this key comment. Yes, we faced problems in getting literature regarding COVID-19 among taxi drivers or other drivers of cars. Thus, we used several studies in different source populations. Despite we used these studies for discussions, we undertook precautions when making such comparisons, and tried to making general statements. Please see the updated discussion sections from page 13 to 16. 

10. For Table 2, is it divided into several sub-tables or it was just a numbering error?

Response: It is a separate Table, which is Table 2, but we found that the title of the table was confusing. We updated the title of the table as “Knowledge and attitude status about COVID-19 among taxi drivers in Dessie City and Kombolcha Town, Northeastern Ethiopia, July to August 2020”. Please see Table 2. 

Reviewer # 2. Review report 

The manuscript reports the findings regarding “Proportion of facemask wearing to prevent COVID-19 and associated factors among taxi drivers in urban areas of Dessie City and Kombolcha Towns in Ethiopia”. This kind of research topic is much relevant, timely and worrying much in the context of LMICs like Ethiopia and the pandemic today’s world as a whole. The analysis was well done. However, I have some concerns (on heading, abstract, methodology, and result sections) which needed be addressed to get published in PLoS ONE. Author(s) can also refer to the attached pdf file.

Response: We thank the reviewer for the positive reflection of our work. We accepted almost all the comments and we amend the paper accordingly. We also referred the PDF version comments for revision. We really appreciate your effort. 

• Authors did not give line pages to make review work ease. Why??

Response: We are sorry that we missed to give line numbers during the original submission. Now we gave lines and please see the revised version. 

Heading of the study:

• Authors need to the word Facemask, two words (face mark), not one word as author(s) described it in the title (not facemark). Keep the consistency across the manuscript.

Response: We thank the reviewer for this comment. However, most commonly, facemask is written as one word and we used as one word through the paper (See the revised versions). 

• Authors are calling Dessie as a city. I suggest they need to use both Dessie and Komblcha as a town because both are really towns. In the current status, Dessie and her sister other urban centers like Debre Markos, Nekemte, etc do not qualify for the standard of to be named as city (such hierarchy). Even regional capitals do not officially maintain such a rank except Addis. 

Response: Although we acknowledge your concern, in Amhara Region, Dessie, Bahir Dar and Gondar are under the category of metro Politian and name as city, whereas others urban areas in Amhara region called as towns like Kombolcha, Debre Markose. This is due to the large size nature of the population. The fact also, Kombolcha is very small compared to Dessie in terms of population size. So, saying the name City and town is based on the government category. 

Abstract

• The last sentence of the “Background” section under the Abstract is too long sentence and lacks language clarity. It has been rewritten.

Response: We revised the abstract throughout and we addressed your concern very carefully. Please see the Background” section under the Abstract in page 2 from lines 24 to 30.. 

• The “Conclusion” section under the Abstract which was submitted online in the submission system (first page of the submission system) was not complete. Authors need to recheck.

Response: We revised the manuscript for language clarity and we corrected the errors of the online submission (See the conclusion section under the abstract and online system). . 

Introduction:

• PLoS ONE use Vancouver styles of citations, use series Arabic numerals or number citations inside a rectangular shaped bracket, [ ]. For further insight, get and read in-text-citation styles and bibliography writing from authors guidelines of the Journal. So, they need to rework accordingly.

Response: We thank you for this key comment. We reformatted the references using PLoS ONE guidelines. Please see the revised version reference throughout the paper. 

• Authors try to cite or use international and national figures (reports) in the Introduction section to describe the background picture of the study topic. In this regard it is well done. However, I have some concerns. The introduction part needs more detail review of relevant empirical works the world experienced on pandemic diseases like COVID-19 in different time. I suggest that authors improve the description in this regard to provide more justification for their study, specifically, they should expand upon the knowledge gap being filled citing adequate related works.

Response: We duly appreciate this insight of the reviewer to improve the background section of the paper. We reviewed the recent 21st century world experience on the pandemic disease at different times. Please see the revised version in lines in page 4 lines 68 to 78. 

Methodology: 

• Under study design and study areas, Why opt for Dessie and Kombolcha towns than other higher urban center in the region? Here, author(s) needs to justify their rationale. In the same section, authors stated something contradicting with what they reported in the " Abstract" section under the "Method". There in the Abstract section and in each Table caption, they mentioned that the study was conducted June to August 2020 whereas here they stated it was conducted from July to August. Keep the consistency across the manuscript. 

Response: The study was conducted during July and August and we did consistency throughout the paper. The study area was Dessie City and Kombolcha town and in the study area section we mentioned the two urban areas specifically since saying urban center did not specifically address these areas. For reasoning of City and Town naming, we already explained above. Thank you. 

• Authors also said like this, the town lies on the intersection of 11°8′N 39°38′E. this shows a point, not an area. I hope, Dessie is a polygon made up of two or more kebeles. They need to use both the latitudinal and longitudinal extents of the two urban centers independently to depict their astronomical location because these towns are here polygons (areas) since they are study settings, should not be a point in such a study. I also advise authors to show the location of the study areas mapping the two study settings using GIS environment.

Response: Thank you for this pertinent comment. We deleted the town lies on the intersection of 11°8′N 39°38′E since your idea is correct. We understand the value of Map to make easy of identification of the study area location to the readers; however, it is less relevance for our study and maps are very important when the study was a community-based study that collect data from house-to-house rather than taxi drivers.

• Under Methodological part, there is a sub-section called “Outcome variable measurement and estimation of proportion of facemask wearing” I have found this miss placed sub heading, no need of stating it here as a method of data analysis because it is expected (even you described it in your method of Data magt and analysis sub-section) that authors undertake descriptive statistics like proportion, cross tabs, and like that below. avoid such repetition of data analysis from here. So, advise author(s) move it (this sub-section) immediate to the "Result" section and reportedly presents the two prevalence, wearing and not wearing face cover. authors also gave definition for Proportion of facemask wearing under Operational definitions below. Likewise edit the coding of outcome variables like this, … otherwise, those who do not wear the face mask was coded 0.

Response: We totally accepted this comment. Thus, we revised based on the suggestions and moved outcome variable measurement and estimation of proportion of facemask wearing section to the data management and analysis section. Please see the revised version in pages 9 from lines 208 to 221. Thank you very much for your insight, 

Ethics approval and consent to participate

• Cite a protocol code that verified approval IRB of the WU/college

Response: We included the IRB protocol number and also we updated the ethical consideration (See in page 10 from lines 224 to 233).

In text citation

• PLoS ONE use Vancouver styles of citations, use series Arabic numerals or number citations inside a rectangular shaped bracket. read in-text-citation and bibliography citation styles of authors guideline. So, rework accordingly

Response: We agreed with this comment and we formatted the reference using the PLoS ONE author guideline and please see the revised version. 

We would like to thank the reviewers and editors for evaluating our manuscript. We have tried to address all the concerns in a proper way and believe that our paper has been improved considerably. We would be happy to make further corrections if necessary and look forward to hearing from you all soon. 

I hope that the revised manuscript is accepted for publication in PLoS ONE. 

Sincerely yours,

Metadel Adane (PhD in Water and Public Health)

---

## [Decision Letter · Decision Letter 1]

10 Feb 2021

PONE-D-20-33269R1

Proportion of taxi drivers wearing facemask to prevent COVID-19 transmission and associated factors in urban areas of Dessie City and Kombolcha Town in Ethiopia

PLOS ONE

Dear Dr. Adane (PhD),

Thank you for submitting your manuscript to PLOS ONE. After careful consideration, we feel that it has merit but does not fully meet PLOS ONE’s publication criteria as it currently stands. Therefore, we invite you to submit a revised version of the manuscript that addresses the points raised during the review process.

The manuscript has greatly improved from the original one. However, English language still need to be paid attention to. I recommend that you seek the help from someone offering professional English editing services.

In addition, the titles of tables and Figures should be revised. Table 1 title should read " Bivariate analysis of socio-demographic and economic factors associated with facemask wearing among .........."

Table 2 title should read " Bivariate analysis of behavioral related factors associated with face mask wearing among  .....". Table 3 should also re-written in the same way. Table 4 title should read " Multivariable analysis of factors associated with ............".

Please do the same correction to the Figure.

In lines 46 to 53, you don't need include the frequency of people who posses a given property (e.g. those who wore masks). Just presenting the their percentage (relative frequency)  is enough as the frequency can be calculated from the total when the percentage is given. This will also make it easy to read these sentences.

Further more, please avoid including the actual results in the discussion.

We look forward to receiving your revised manuscript.

Kind regards,

Martin Chtolongo Simuunza, PhD

Academic Editor

PLOS ONE

Reviewers' comments:

Reviewer's Responses to Questions

**Comments to the Author**

1. If the authors have adequately addressed your comments raised in a previous round of review and you feel that this manuscript is now acceptable for publication, you may indicate that here to bypass the “Comments to the Author” section, enter your conflict of interest statement in the “Confidential to Editor” section, and submit your "Accept" recommendation.

Reviewer #1: All comments have been addressed

2. Is the manuscript technically sound, and do the data support the conclusions?

Reviewer #1: Yes

3. Has the statistical analysis been performed appropriately and rigorously? 

Reviewer #1: Yes

4. Have the authors made all data underlying the findings in their manuscript fully available?

Reviewer #1: Yes

5. Is the manuscript presented in an intelligible fashion and written in standard English?

Reviewer #1: Yes

6. Review Comments to the Author

Reviewer #1: Author(s) have well addressed all concerns and comments as per of reviewer's suggestions. In this submission, a reviewer declares no more comments or concerns. Hence, he recommends "accepted" for final production. Finally, a reviewer would like to congratulate all the authors once again and wish them the best.

7. PLOS authors have the option to publish the peer review history of their article (what does this mean?). If published, this will include your full peer review and any attached files.

Reviewer #1: **Yes: **Habtamu Tolera

---

## [Author Response · Author response to Decision Letter 1]

16 Feb 2021

Dear Editor

Thank you for your letter with a decision of minor revision needed. We agree with all the comments/questions raised and please find for each below. 

Response to the editor comment: 

The manuscript has greatly improved from the original one. However, English language still need to be paid attention to. I recommend that you seek the help from someone offering professional English editing services.

Response: We appreciate your concern and we received English language editing support from the native English speaker. We acknowledged the expert in page 16 in line 376 to 377 of the acknowledgment section. I marked areas within the red color that language improvement done throughout the manuscript. Please see also the areas that language improvement done. See the red marked section of the revised manuscript. 

In addition, the titles of tables and Figures should be revised. Table 1 title should read " Bivariate analysis of socio-demographic and economic factors associated with facemask wearing among .........."/ Table 2 title should read " Bivariate analysis of behavioral related factors associated with face mask wearing among .....". Table 3 should also re-written in the same way. Table 4 title should read " Multivariable analysis of factors associated with ............".

Response: We accepted all your suggestions for the titles. Please see the titles of Table 1, 2, 3 and 4. Thank you. 

Please do the same correction to the Figure.

Response: The title of Figure 1 is ok as it is (Please see title of Figure 1). 

In lines 46 to 53, you don't need include the frequency of people who posses a given property (e.g. those who wore masks). Just presenting the their percentage (relative frequency) is enough as the frequency can be calculated from the total when the percentage is given. This will also make it easy to read these sentences.

Response: Yes, very crucial comment. We updated the manuscript as per your suggestion please see the revised version in the abstract in page 3 line 47 to 54. See also in the result section in pages from 11 to 12 marked with red color. 

I hope that the revised manuscript is accepted for publication in PLoS ONE. 

Sincerely yours,

Metadel Adane (PhD) 

Wollo University, Department of Environmental Health

---

## [Editor Report · Decision Letter 2]

17 Feb 2021

Facemask wearing to prevent COVID-19 transmission and associated factors among taxi drivers in Dessie City and Kombolcha Town in Ethiopia

PONE-D-20-33269R2

Dear Dr. Adane (PhD),

We’re pleased to inform you that your manuscript has been judged scientifically suitable for publication and will be formally accepted for publication once it meets all outstanding technical requirements.

Kind regards,

Martin Chtolongo Simuunza, PhD

Academic Editor

PLOS ONE
---

## [Editor Report · Acceptance letter]

24 Feb 2021

PONE-D-20-33269R2 

Facemask wearing to prevent COVID-19 transmission and associated factors among taxi drivers in Dessie City and Kombolcha Town, Ethiopia 

Dear Dr. Adane:

I'm pleased to inform you that your manuscript has been deemed suitable for publication in PLOS ONE. Congratulations! Your manuscript is now with our production department. 

Kind regards, 

on behalf of

Dr. Martin Chtolongo Simuunza 

Academic Editor

PLOS ONE